# OpenReview forum: "DiffuGuard: How Intrinsic Safety is Lost and Found in Diffusion Large Language Models"
_ICLR.cc/2026/Conference — ICLR 2026 Poster_

### Official Review · Reviewer_3R1N · 2025-10-28

**Soundness:** 3
**Presentation:** 3
**Contribution:** 3
**Rating:** 6
**Confidence:** 3

**Summary:**

This paper analyzes safety vulnerabilities specific to DLLMs, stemming from their parallel generation and iterative refinement mechanisms. The authors identify two primary issues: a harmful bias in the standard greedy remasking strategy at the intra-step level, and a "Denoising-path Dependence" at the inter-step level, where the safety of early-generated tokens heavily influences the final output. To mitigate these issues, the paper introduces DIFFUGUARD, a training-free defense framework. DIFFUGUARD employs Stochastic Annealing Remasking to introduce controlled randomness into the remasking process and Block-level Audit and Repair to autonomously detect and correct unsafe content using internal representations. Experimental results on four dLLMs show this framework reduces the average Attack Success Rate (ASR) from 47.9% to 14.7% across six attack methods , with a reported minor impact on model utility and inference latency.

**Strengths:**

This paper tackles the problem of safety in Diffusion Large Language Models, which I think is a relevant and underexplored area compared to autoregressive models. The paper provides an analysis of vulnerabilities it claims are specific to dLLMs. The authors frame this analysis around "intra-step" and "inter-step" dynamics, which is a conceptualization that helps to structure the problem. A central part of their analysis is the concept of "Denoising-path Dependence", which they propose as a key difference in how dLLMs' safety can be compromised.



Regarding the technical approach, the proposed defense, DIFFUGUARD, is directly motivated by the preceding analysis. Its two components, Stochastic Annealing Remasking and Block-level Audit and Repair, are designed to map onto the two identified vulnerabilities (intra-step and inter-step, respectively). I find this link between the problem analysis and the solution design to be logical. The experimental validation covers four dLLMs and six different attack methods , including some dLLM-specific attacks like PAD and DIJA. The authors also include experiments on model utility and inference speed , which is important for assessing the method's practical viability as a "plug-and-play" module. The paper is generally clearly written, and figures like Figure 1 are helpful for understanding the overall framework.

**Weaknesses:**

My primary concern is with the evaluation methodology, which I believe overstates the robustness of the proposed defense. The experiments are conducted against a set of *static* and known attack methods. However, the defense itself, particularly the "Block-level Audit and Repair" component, is a fixed, rule-based mechanism. **A large body of recent work [1, 2, 3] has demonstrated that such static defenses are highly vulnerable. An "adaptive attacker" who has knowledge of the defense's strategy can almost always find a way to bypass it, often reducing the defense's effectiveness to near 0%**[1, 2, 3].

I am highly skeptical that the reported 14.7% ASR would hold under such an attack. For example, an attacker could almost certainly use gradient-based methods to optimize a jailbreak prompt that *also* minimizes the "Safety Divergence" (SD), ensuring it stays below the detection threshold $\lambda$. This would render the entire "Audit and Repair" module useless. While I personally find the idea that diffusion models may have some inherent robustness plausible [2] (where they provide certified lower bounds for two diffusion kernels) , the robustness claimed *by this specific defense* is, in my opinion, an artifact of a weak threat model. **I would strongly urge the authors to re-evaluate their framework against a simple, adaptive, white-box attack that directly targets the SD metric, and using the attacks in Carlini's recent paper [1]**.

Beyond this central point, I have a few other concerns. The 'Block-level Audit' mechanism, for instance, relies on a 'core hypothesis' that a jailbreak prompt can be decomposed into a malicious core $p_{origin}$ and an adversarial template. This seems plausible for attacks like PAD or DIJA, but it's unclear how $p_{origin}$ would be automatically extracted from more complex, natural-language jailbreaks, such as the WildJailbreak example in Figure 8. While the appendix provides a good sensitivity analysis for hyperparameters like $\alpha_0$ and $\lambda$, the paper doesn't describe the *process* for selecting the final values used in the main experiments, like $\lambda=0.9$. For reproducibility, it would be helpful to know how these values were chosen and how to best tune them for new models. The evaluation is also focused on the Masked Diffusion Model (MDM) paradigm. It would be interesting to know if the authors believe these same vulnerabilities and the DIFFUGUARD defense would apply to other dLLM architectures, such as the uniform kenerls. Finally, the main ablation study shows both *modules* are necessary, but a more fine-grained ablation on the *sub-components* would be even more convincing. For example, in the first module, it would be useful to see a direct comparison of a greedy, a constant-stochastic, and the proposed annealed-stochastic strategy to fully isolate the benefit of the annealing schedule itself.

---
**References**

[1] Carlini et al. (2024). The Attacker Moves Second: Stronger Adaptive Attacks Bypass Defenses Against Llm Jailbreaks and Prompt Injections.
[2] Chen et al. (2024). Towards the worst-case robustness of large language models.
[3] Andriushchenko et al. (2024). Jailbreaking Leading Safety-Aligned LLMs with Simple Adaptive Attacks.

**Questions:**

My primary question relates to the robustness of the defense under a stronger, adaptive threat model.

1.  Your evaluation is conducted against a set of *static* and known attack methods. However, a large body of recent work [1, 2, 3] has shown that static, rule-based defenses often provide a false sense of security and their effectiveness drops significantly (often to near 0%) when evaluated against an *adaptive attacker* who knows the defense mechanism. The "Block-level Audit" module, which relies on a fixed "Safety Divergence" (SD) threshold $\lambda$, seems particularly vulnerable to a simple, white-box attack that uses gradients to find a prompt that is both harmful *and* has an SD value below $\lambda$. Could you please comment on this limitation? I would ask that you perform an evaluation against such an adaptive attack, as this would be critical for assessing the true robustness of DIFFUGUARD.

---
**References**

[1] Carlini et al. (2024). The Attacker Moves Second: Stronger Adaptive Attacks Bypass Defenses Against Llm Jailbreaks and Prompt Injections.

---

> ### Author Response · Authors · 2025-11-21
> **Responses to Reviewer 3R1N (1/2)**
>
> Dear **Reviewer 3R1N**:
>
> We sincerely thank you for your positive feedback and valuable suggestions on our work! We are pleased to clarify and answer the questions and provide some additional experimental results.
>
> # For Weakness 1
>
> Thank you for your insight.
>
> We respectfully disagree that the 'Block-level Audit and Repair' is a fixed, rule-based mechanism. On the contrary, it incorporates significant stochasticity. During the Repair phase, we randomly remask tokens within the block according to a certain probability and then regenerate the block over a few steps. If an attacker, aware of our mechanism, were to train a corresponding adversarial attack (e.g., an adapted GCG), the stochasticity we introduce would cause a deviation in the attack optimization, leading to its failure.
>
> We have also additionally evaluated the effectiveness of several adaptive attack methods, as shown in the table below:
>
> | ASR(%)     | PAD$_\text{AdvBench}$ | + Multi-sampling Attack | + Gradient-based Attack (on SD) | + Heuristic threshold-probing Attack (on SD) |
> | ---------- | --------------------- | ----------------------- | ------------------------------- | -------------------------------------------- |
> | LLaDA-8B   | 93.65                 | 98.85                   | 88.27                           | 98.84                                        |
> | DiffuGuard | 24.42                 | 35.96                   | 44.42                           | 65.58                                        |
>
> **We have included the detailed setup and results of these adaptive attacks in Appendix D.3.**
>
> # For Weakness 2 & Question 1
>
> Thank you for your insightful point. While it is true that an attacker could attempt to minimize the Safety Divergence (SD) via optimization, we argue that successfully optimizing such a jailbreak prompt is exceptionally difficult.
>
> Our rationale is that SD, by definition, measures the representational difference between the model's understanding of the original malicious prompt $p_\text{origin}$ (which is likely perceived as dangerous, requiring refusal) and its understanding of the optimized jailbreak prompt $p_0$ (which is designed to appear safe, requiring a response). This creates an intrinsic optimization paradox for the attacker. To successfully jailbreak the model, the attacker must optimize the prompt *away* from the dangerous $p_\text{origin}$ and towards a benign-looking state, an action that, by its very nature, maximizes the SD. Conversely, to evade detection by DiffuGuard's Audit module, the attacker must *minimize* the SD, which would require optimizing the prompt back *towards* the original dangerous $p_\text{origin}$ state.
>
> We posit that these two objectives are contradictory. The loss functions for a successful attack and for evading our defense are inherently in conflict, making the joint optimization highly challenging.
>
> To demonstrate this, we conducted an experiment using a gradient-based, white-box attack that directly targets the SD metric. The experimental results are included in our response to Weakness 1. As the results show, this type of attack, which directly optimizes against the SD metric, is not effective against DiffuGuard, thus demonstrating the robustness of our framework.
>
> **We have added the above discussion and the experimental results to the final version of the paper.** Thank you again for your perspective.

---

> ### Author Response · Authors · 2025-11-21
> **Responses to Reviewer 3R1N (2/2)**
>
> # For Weakness 3
>
> Your concerns are reasonable.
>
> 1. **Regarding the extraction of $p_\text{origin}$:** DIJA and PAD are currently the primary attack methods targeting dLLMs. They leverage the in-place prompting capability of dLLMs, using templates to guide the model toward generating harmful answers. We believe this is a unique and highly effective attack method specifically for dLLMs. The main contribution of DiffuGuard is proposing this general idea for detecting and defending against jailbreaks. If attack methods are optimized, DiffuGuard can also be optimized. If more complex and effective in-place prompting jailbreaks emerge in the future, such as those hiding templates in natural language, we believe DiffuGuard would not be difficult to adapt—for instance, by using rule-based token segment splitting or by invoking an LLM for semantic analysis and decomposition.
>
> 2. **Regarding hyperparameters:** As you noted, we presented detailed hyperparameter analysis in Appendix D. As for the selection process, we conducted experiments across many hyperparameter configurations and found that DiffuGuard consistently improved safety while maintaining general capabilities under various settings. Therefore, we selected a set of reasonable configurations for each model to present our main results.
>
> 	1. $\alpha_0$: As discussed in Section 3.2, a high $\alpha_0$ enhances safety but potentially degrades general capability. For new models, we recommend starting at 0.5 and tuning it to achieve the desired balance between safety and utility.
>
> 	2. $\lambda$ and $\gamma$: These parameters function together in the Block-level Audit and Repair module. $\lambda$ is a model-specific parameter, as different models' hidden states exhibit varying sensitivities to cosine similarity. $\gamma$ also represents a trade-off between safety and general capability.
> 	3. extra_steps: We set this to 8, which we found to be an efficient value that does not significantly impair model utility.
>
> 3. **Regarding other dLLM architectures:** dLLM is a broad term, and its specific implementations vary significantly. Our paper focuses exclusively on Masked Diffusion Models, which have recently demonstrated performance comparable to AR LLMs. However, we also believe that other dLLM paradigms (such as continuous dllm, flow-matching, and uniform kernels you mentioned) are likely to exhibit some similar safety phenomena, stemming from the generic parallel generation and iterative refinement process common to dLLMs.
>
> 4. **Regarding the fine-grained ablation study:** We agree with your idea. We have conducted more fine-grained ablation experiments targeting the sub-components within each module, as shown in Table 2 in our revised paper.
>
> **We have revised the paper to include these results.** Thank you again for your suggestions!
>
> ---
>
> **We sincerely look forward to further discussion to clarify any other concerns. If you have additional suggestions, we are committed to addressing them and continuously improving our work based on your feedback.**

---

### Official Review · Reviewer_oL9w · 2025-10-28

**Soundness:** 1
**Presentation:** 2
**Contribution:** 2
**Rating:** 2
**Confidence:** 4

**Summary:**

The authors propose DiffuGuard, a defense framework for Diffusion Large Language Models. By first performing an analysis of model vulnerabilities, focusing on the specific weaknesses of the model’s decoding process, the authors design the framework to mitigate the generation of harmful responses through modified remasking and repair mechanisms. The authors evaluate the defense framework with several jailbreak methods on 4 models, reporting how DiffuGuard is able to decrease the attack success rate while maintaining model utility and efficiency.

**Strengths:**

- Section 3 offers a good perspective of dLLMs' security issues: the analysis of the model’s decoding mechanism vulnerabilities is clear and well presented.
- The authors evaluate the defense against dLLM-specific jailbreak methods.

**Weaknesses:**

- The Block Level Audit process may not function as intended.
- Several experimental setup choices limit the contribution of the work.

**Questions:**

- My main concern lies in the block-level audit procedure. The authors draw inspiration from[1], who identify a refusal direction by contrasting “harmful” and “harmless” representations. Similarly, the authors compute the difference between the representations of a malicious request and its corresponding jailbreak version (same intent but with an adversarial template).

    I see two main issues with this methodology:

    1. It assumes that the model inherently refuses *p_origin*, since this is treated as the safety baseline for computing the safety divergence. However, there is no guarantee that the model is actually aligned to reject the original malicious query, meaning that the baseline representation of *p_origin* may not correspond to a genuine refusal state.
    2. Depending on the jailbreak strategy, different *p_template* components produce different internal representations (*H_p0*). Some templates may shift the representation closer to benign input clusters, while others remain closer to harmful ones [2]. Consequently, the information captured by SD varies across jailbreak types, which may reduce the reliability and effectiveness of the defense. While the approach appears effective against attacks specifically designed for dLLMs (as shown in Table 2), the paper does not include an ablation across diverse jailbreak algorithms such as AutoDAN.
- The authors evaluate DiffuGuard under two attack scenarios:

    – Pre-optimized attacks, which rely on datasets of already computed jailbreak prompts;

    – Online Generative Attacks, where the attack is optimized in real time starting from a malicious query and adapting based on the model’s feedback (e.g., the optimization of an adversarial suffix as in GCG).

    While this distinction is useful, especially for assessing the transferability of pre-optimized prompts to dLLMs, I find the selection of Online Generative Attacks rather limited.
    First, at the beginning of Section 5.2, the authors state: *“we employ 5 mainstream attack algorithms to generate attacks in real-time,”* yet results are reported for only 4.

    Second, GCG, which is included in this category, is implicitly presented as an online attack against the target model. However, Appendix B.3 specifies that all adversarial suffixes are trained on LLaMA-3-8B-Instruct. Beyond the lack of explanation for this particular surrogate choice, I find it misleading to classify this setup as an Online Generative Attack, since it effectively corresponds to a pre-optimized attack later applied to other models.

    Finally, GCG in this configuration appears to be ineffective, yielding 0.0 ASR on three out of four models. This alone undermines the evaluation of the defense, as the attack itself fails to succeed. I would therefore suggest that the authors employ more effective state-of-the-art attack techniques [3][4], which would provide a more complete and reliable assessment of DiffuGuard’s robustness.

- One of the main challenges in the current literature on evaluating attacks and defenses for language models, lies in the methodology used to compute the ASR. Using an arbitrary “LLM-as-a-judge,” with an arbitrary system prompt, makes it difficult to meaningfully compare the paper’s results with those from other works. This is precisely why recent studies have introduced dedicated benchmarks [5][6], featuring models fine-tuned specifically for this evaluation task, allowing for a consistent and reliable comparison across defenses. Consequently, I consider the use of GPT-4o-mini as a strong limitation of the authors’ evaluation setup, as it does not provide a robust or standardized measure of the defense’s effectiveness.
- Although the analysis in Section 3 is insightful, the authors base both their observations and conclusions solely on results from a single model. It would be important to verify whether the same vulnerabilities appear in other diffusion LLMs, and whether these weaknesses generalize across different model architectures.

Minor points:

– In Figure 1, some elements are difficult to read, especially on the right side.

– In Table 2, it would be useful to also report the contribution of each component of the defense on *safe queries*.

– For in Section 3, Random Remasking, there may be degradation of model’s utility*,* the authors acknowledge this issue when considering jailbreak inputs. it would be useful to  analyze also how much performance drops on safe queries.

– In Figure 4, it would be interesting to include the effect of a jailbreak query with the prefilling token *“Sure”* to highlight differences compared to the *“Sorry”* prefilling.

References

[1] Arditi et al., Refusal in Language Models Is Mediated by a Single Direction, 2024

[2] Ball et al., Understanding Jailbreak Success: A Study of Latent Space Dynamics in Large Language Models, 2024

[3] Mazeika et al., HarmBench: A Standardized Evaluation Framework for Automated Red Teaming and Robust Refusal, 2024

[4] Souly et al., A StrongREJECT for Empty Jailbreaks, 2024

[5] Andriushchenko et al., Jailbreaking Leading Safety-Aligned LLMs with Simple Adaptive Attacks, 2025

[6] Mehrotra et al, Tree of Attacks: Jailbreaking Black-Box LLMs Automatically, 2024

---

> ### Author Response · Authors · 2025-11-21
> **Responses to Reviewer oL9w (1/2)**
>
> Dear **Reviewer oL9w**:
>
> We sincerely thank you for your thoughtful feedback and valuable suggestions on our work! We are pleased to clarify and answer the questions and provide some additional experimental results.
>
> # For Weakness 1 & Question 1
>
> Thank you for your in-depth feedback.
>
> 1. Regarding your first concern: We acknowledge that our method is based on the assumption that the model inherently refuses $p_\text{origin}$, and we believe this assumption is reasonable. As we investigated in our experiments in Section 3.3, dLLMs exhibit a perfect refusal rate for standard Malicious Queries (We have validated this across other dLLMs). This indicates that dLLMs do possess a reliable inherent safety mechanism capable of rejecting such harmful queries, which validates the use of $h_\text{origin}$ as a safety baseline.
> 2. Regarding your second concern: We concur with your point that different jailbreak methods may steer the input representation towards different clusters, as revealed in [2]. However, we argue that this observation also supports our approach. Figure 7 in [2] demonstrates that all effective jailbreak attacks, despite their differences, function by suppressing the 'harmfulness' feature. We believe this strongly supports the rationale for our Safety Divergence. Essentially, SD is designed to quantify this exact phenomenon: it identifies a jailbreak attempt by measuring the significant deviation from the inherent 'harmfulness' representation ($h_\text{origin}$) that occurs when a template forces the model to masquerade as benign. As for the ablation study on more diverse jailbreak algorithms like AutoDAN, we agree this is a valuable suggestion and have added it to our Table 2.
>
> **We have updated our paper to include more fine-grained ablation experiments with a wider range of attack methods (see Table 2).** We sincerely thank you for your perspective, as it is highly valuable for improving our work.
>
> [1] Arditi et al., Refusal in Language Models Is Mediated by a Single Direction, 2024
>
> [2] Ball et al., Understanding Jailbreak Success: A Study of Latent Space Dynamics in Large Language Models, 2024
>
> # For Weakness 2 & Question 2
>
> Thank you for your insights.
>
> First, we confirm there was a typo in Section 5.2. It should indeed state, "we employ **3** mainstream attack algorithms..."(if GCG is excluded). We appreciate you pointing this out.
>
> Regarding the online generative attacks, we acknowledge the limitations in our selection. Our survey revealed a scarcity of online attack options specifically for dLLMs, largely due to their distinct generation paradigm compared to AR LLMs. At the time of the conference deadline, no effective GCG-style attack strategy specifically for dLLMs had been established. Consequently, we relied on the known transferability of GCG and used prompts pre-optimized on LLaMA-3-8B-Instruct to attack the dLLMs.
>
> We concede that classifying this setup as an 'Online Generative Attack' was misleading, and **we have corrected this categorization in the revised paper.** As suggested by other reviewers, **we have also conducted extra evaluations against adaptive attacks, which have been added to the paper (you can refer to Appendix D.3).**
>
> We have adopted your suggestion to re-evaluate the Attack Success Rate using HarmBench[3] and StrongReject[4]. Results are shown in the following table.
>
> |            | HarmBench(ASR) |         |       | StrongREJECT(Score) |                       |                        |
> | ---------- | -------------------- | ------- | ----- | ------------------------- | --------------------- | ---------------------- |
> |            | GCG                  | AutoDAN | PAD   | WildJailbreak             | PAD$_\text{AdvBench}$ | PAD$_\text{HarmBench}$ |
> | LLaDA      | 2.23%                | 0.99%   | 1.24% | 0.0928                    | 0.7064                | 0.3467                 |
> | DiffuGuard | 0.74%                | 0.74%   | 0.50% | 0.0425                    | 0.1771                | 0.1417                 |
> | Dream      | 0.25%                | 0.00%   | 0.00% | 0.0169                    | 0.6343                | 0.0275                 |
> | DiffuGuard | 0.00%                | 0.00%   | 0.00% | 0.0092                    | 0.0994                | 0.0269                 |
>
> It is evident that HarmBench and StrongReject utilize a more stringent jailbreak evaluation methodology. Under this demanding assessment setting, DiffuGuard continues to exhibit robust defense capabilities.
>
> **We have updated the paper with these new results, you can find them in Appendix D.5.** Thank you again for your valuable suggestion!

---

> > ### Comment · Reviewer_oL9w · 2025-11-25
> >
> > Thanks for your detailed answer!
> >
> > I would like to discuss a few points:
> >
> > **W1**:  I understand your point. I see that in Figure 4, Section 3.3, you report the ASR of LLaDA-8B-Instruct against the malicious queries (0%), validating the assumption that p_origin is refused. The same behavior applies to Dream-7B in Appendix D.4.
> > What I'm worried about is what happens when this assumption does not hold: what if we want to protect a model that is not fully aligned to refuse the initial malicious query? Will DiffuGuard fail in this kind of scenario?
> > Although the assumption is validated for two models (only on the malicious query that the authors considered), I still see this assumption as a limitation of the approach.
> >
> > **W2**: Thank you for acknowledging the wrong categorization for GCG. I still do not understand whether there is a reason why you considered GCG jailbreaks against Llama3-8B-Instruct in your evaluation or not. You mention you _"relied on the known transferability of GCG "_: here, you mean GCG in general or the transferability of GCG optimized on Llama3-8b? If you mean the latter, could you provide a reference?
> > Regarding the re-evaluation of ASR values. I believe the values you reported underline the limitations of the setup you used to evaluate the effectiveness of DiffuGuard.
> > I see two main problems:
> > 1. *High Discrepancy between ASR judges*:
> > Let's consider AutoDAN's and PAD's ASR against Llada.
> > According to Table 1, the results you reported with gpt-4o-mini as a judge are 39.23% and 93.65% respectively. With the Harmbench classifier are 0.99% and 1.24%. Is this drastic decrease due to the fact that you are changing the prompts (from AdvBench to HarmBench) or simply due to the fact that the HB classifier is more restrictive?
> > If you changed the prompts, I would suggest reporting the ASR values with the HarmBench classifier on the same dataset to have a fair comparison. If the prompts are the same, I'm really worried about the reliability of the judge you considered for the ASR computations reported in Table 1.
> >
> > 2. *The attacks you are considering are not working on the base model*:
> > Generally, what I see from these results is that the attacks you are considering are not effective against the models without defenses. How could you evaluate a defense contribution if the attacks you are considering are not effective?
> > According to Table 1, GCG achieves a 0% ASR on 3 out of 4 models. Considering the HarmBench ASR, which you reported, AutoDAN and PAD also achieve a very low ASR. This could also stem from an insufficient optimization budget considered for these attacks. Although the other attacks you considered, whose values are reported in Table 1, achieve high ASR, I find those values somehow invalidated, due to the high discrepancy between the ASR computed with the judge you considered and the HB one.

---

> ### Author Response · Authors · 2025-11-21
> **Responses to Reviewer oL9w (2/2)**
>
> # For Question 3
>
> Thank you for this insightful point. We agree with your perspective that using a unified, standard evaluation method enhances the reliability and reproducibility of experimental results.
>
> As you suggested, we have conducted additional evaluations using: 1) LLaMA Guard, and 2) Human evaluation (due to time and resource constraints, we sampled 200 items for each jailbreak method for this analysis). Our results are presented in the table below.
>
> |            | WildJailbreak |                             |                    | PAD_AdvBench |                             |                    |
> | ---------- | ------------- | --------------------------- | ------------------ | ------------ | --------------------------- | ------------------ |
> |            | LLaMA-Guard   | Human | gpt-4o-mini (Ours) | LLaMA-Guard  | Human | gpt-4o-mini (Ours) |
> | LLaDA      | 7.75          | 22.50                       | 23.95              | 86.54        | 91.50                       | 93.65              |
> | DiffuGuard | 2.45          | 6.00                        | 8.50               | 17.12        | 21.50                       | 24.42              |
> | Dream      | 0.40          | 2.00                        | 3.30               | 70.38        | 90.00                       | 99.23              |
> | DiffuGuard | 0.10          | 0.50                        | 1.05               | 19.42        | 30.50                       | 37.31              |
>
> As the results show, although LLaMA Guard is stringent in its criteria for judging attack success, it exhibits similar trends to GPT-4o-mini assessment. Furthermore, our GPT-4o-mini assessment demonstrates strong alignment with the Human Expert evaluation.
>
> We have merged these comparative results into the main tables in the revised paper. Thank you for this valuable suggestion!
>
> # For Question 4
>
> We agree that verifying whether these conclusions generalize by analyzing other dLLMs is a very important point.
>
> **We have updated our paper to include our analysis on other dLLMs in Appendix D.4.** Thank you for your suggestion!
>
> # For Minors
>
> - Thank you for pointing this out. We have optimized the figure to make it easier to read.
> - We have added a more fine-grained ablation study to Table 2, targeting each component of the defense.
> - We have added the impact of Random Remasking on Safe Queries (we use GSM8K) to Table 2.
> - Thank you for the suggestion. We found that adding "Sure" to the jailbreak query increases the ASR to 94.7%. We have updated Figure 4 accordingly.
>
> ---
>
> We sincerely look forward to further discussion to clarify any other concerns. If you have additional suggestions, we are committed to addressing them and continuously improving our work based on your feedback. **If you are satisfied with our rebuttal and the revisions, we kindly hope you will consider increasing our score.**

---

> > ### Comment · Reviewer_oL9w · 2025-11-25
> >
> > I have an additional question regarding the human evaluation.
> > How many people were involved in the labelling of the model's responses?

---

> > > ### Author Response · Authors · 2025-11-25
> > > **Response to Follow-up Questions (1/2)**
> > >
> > > Dear **Reviewer oL9w**:
> > >
> > > Thank you for your time and effort in reviewing our paper. We are pleased to engage in a deeper discussion with you to clarify your concerns.
> > >
> > > # For W1
> > >
> > > Thank you for raising this point. As you stated, this assumption is valid within our current context. LLaDA and Dream are the two most common base models in the dLLM domain, collectively covering the vast majority of existing work. We have conducted detailed experiments on both models to validate the reasonableness of this assumption.
> > >
> > > For "not fully aligned" models, we do not believe they constitutes a practically relevant scenario. Most base models used in LLM safety research are already safety-aligned or have undergone fine-tuning steps, e.g., LLaMA, Qwen, Mistral series. We do not consider models that have received no safety adjustments to be within the scope of practical application, as discussing their safety lacks real-world significance. Such models require safety alignment or instruction fine-tuning before they can be deployed.
> > >
> > > Furthermore, should a scenario necessitate defending a not fully aligned dLLM, our Stochastic Annealing Remasking module can still provide a degree of defense.
> > >
> > > In summary, we maintain that our core assumption is reasonable because it covers the vast majority of practically applicable models in the dLLM domain.
> > >
> > > # For W2
> > >
> > > According to Section 3.2 of [1], GCG attacks possess transferability, with suffixes trained on Vicuna even effectively jailbreaking closed-source models like GPT3.5. Furthermore, since LLaDA shares an architectural similarity with LLaMA (Appendix B.2 of [2]), we chose GCG suffixes optimized on LLaMA-3-8B as the surrogate.
> > >
> > > We acknowledge that the directly transferred GCG showed limited capability and was indeed a weak attack in this specific scenario. This is because currently, no effective GCG method specific to dLLMs exists, and the original GCG's induction logic (inducing "Sure" at the beginning of the response) is fundamentally incompatible with the dLLM's non-fixed generation order.
> > >
> > > To address the limitations of this weak baseline, we supplemented our evaluation by adding an experiment in Appendix D.3 that measures the feasibility of a Gradient-based attack method specifically designed for dLLMs.
> > >
> > > [1] Zou, Andy, et al. "Universal and transferable adversarial attacks on aligned language models." *arXiv preprint arXiv:2307.15043* (2023).
> > >
> > > [2] Nie, Shen, et al. "Large language diffusion models." *arXiv preprint arXiv:2502.09992* (2025).
> > >
> > > # For W2.1
> > >
> > > Thank you for your careful observation. We confirm that we did not change the prompts or the dataset for these evaluations.
> > >
> > > Upon investigation, we found that this discrepancy stems from an issue in HarmBench's default evaluation procedure. The standard HarmBench pipeline uses the `HarmBench-Llama-2-13b-cls` model for safety classification. The default method extracts **only the first word** (yes/no) of the classifier's output to determine safety. However, we found this to be inaccurate, as the crucial 'yes/no' token **sometimes appears later** in the classifier's response.
> > >
> > > To correct this, we re-evaluated the three attacks using HarmBench, extracting the first few words of the classifier's output and applying fuzzy matching to improve accuracy. Our new results are presented in the table below. **We have also revised our paper to update these results.**
> > >
> > > |            | HarmBench(ASR) |         |        |
> > > | ---------- | -------------------- | ------- | ------ |
> > > |            | GCG                  | AutoDAN | PAD    |
> > > | LLaDA      | 7.94%                | 20.84%  | 50.37% |
> > > | DiffuGuard | 2.48%                | 17.37%  | 14.39% |
> > >
> > > Regarding the remaining ASR discrepancy between HarmBench and our original human/LLM-based evaluation, we attribute this to a fundamental difference in safety criteria. The HarmBench classifier labels responses as safe even if they "accept the dangerous prompt but do not produce obviously harmful behavior." Conversely, following the stricter evaluation standard used by DIJA and PAD, we consider such compliant responses to be dangerous.

---

> > > > ### Author Response · Authors · 2025-11-25
> > > > **Response to Follow-up Questions (2/2)**
> > > >
> > > > # For W2.2
> > > >
> > > > Regarding GCG, since no dedicated GCG algorithm currently exists for dLLMs, we relied on directly transferring the adversarial suffix trained on LLaMA-3-8B, which resulted in suboptimal efficacy. Despite the attack's weakness, DiffuGuard still demonstrates a mitigating effect.
> > > >
> > > > As for AutoDAN and PAD, our new HarmBench evaluation results (presented in the table above) validate that their attack efficacy is not critically low. These newly validated baseline attacks confirm that DiffuGuard is capable of effectively defending against prevalent optimization-based and dLLM-specific threats.
> > > >
> > > > # For Human Evaluation
> > > >
> > > > We assigned the human evaluation task to two distinct annotators, who were each responsible for the assessment on the WildJailbreak and PAD_AdvBench datasets, respectively. Their primary task was to perform a binary classification, labeling each response as either safe or unsafe.
> > > >
> > > > Subsequently, one of these annotators performed a final check and re-confirmation across all the assessments to ensure quality assurance.
> > > >
> > > > ---
> > > >
> > > > **We thank you again for the invaluable time spent reviewing our manuscript. Your suggestions are crucial for improving our work. Should you have any further concerns, we would be happy to continue the discussion.**

---

### Official Review · Reviewer_ZfYy · 2025-10-30

**Soundness:** 3
**Presentation:** 2
**Contribution:** 2
**Rating:** 4
**Confidence:** 3

**Summary:**

This paper proposes DiffuGuard, a technique for defending jailbreaks on diffusion LLMs. It first conducts intra-step and inter-step analysis for the dLLM internals and observes two key insights: introducing randomness during the generation can enhance the model safety and early generation steps failure can significantly impact the following generation process. Based on such insights, authors propose DiffuGuard, it first introduces stochastic annealing remasking to inject controlled randomness to balance the safety and utility and conducts a block-level audit and repairs to fix the compromised early generation steps. Evaluation results demonstrate the proposed defense can reduce the jailbreak ASR from 47.9% to 14.7%.

**Strengths:**

- The paper investigates a critical safety issue in diffusion-based LLMs, an emerging and underexplored direction.
- The approach’s motivation is well-supported by in-depth analysis of dLLM internals.
- The evaluation demonstrates the effectiveness of DiffuGuard against state-of-the-art jailbreak attacks.

**Weaknesses:**

- The presentation could be improved; a more intuitive explanation of diffusion LLM mechanisms would help non-experts better grasp the core idea.
- The key observations may not be entirely novel and appear to manifest in AR LLMs as well.
- There is limited discussion of potential adaptive attacks.

**Questions:**

- The two key insights presented are not entirely novel, as similar phenomena have been observed in prior studies on the safety issues of autoregressive-based LLMs. For example, [1, 2] also explored how strategically manipulating the output probability distribution can compromise model alignment. The authors should provide a more detailed discussion to clarify what is unique about their findings compared with existing work. For instance, rather than using the proposed stochastic annealing remasking, could a simple temperature adjustment of the probability distribution during remasking achieve a similar effect? Similarly, regarding the second insight, the observed dependency chain is also a well-known property of autoregressive LLMs, where early tokens play a critical role in maintaining model safety [3].

- I also encourage the authors to refine the presentation, particularly in the background section. The current exposition may be difficult for readers unfamiliar with diffusion-based LLMs to follow. For example, in line 113, the phrase “The tokens at the remaining positions” is ambiguous and requires clarification.

- The evaluation lacks discussion of potential adaptive attacks. For instance, how DiffuGuard performs against stronger adversaries aware of its defense mechanism. Including such experiments would further strengthen the demonstrated robustness of the proposed approach.


---

Reference
--------


[1] Huang Y, Gupta S, Xia M, et al. Catastrophic jailbreak of open-source llms via exploiting generation[J]. arXiv preprint arXiv:2310.06987, 2023.

[2] Zhang Z, Shen G, Tao G, et al. On large language models’ resilience to coercive interrogation[C]//2024 IEEE Symposium on Security and Privacy (SP). IEEE, 2024: 826-844.

[3] Qi X, Panda A, Lyu K, et al. Safety alignment should be made more than just a few tokens deep[J]. arXiv preprint arXiv:2406.05946, 2024.

---

> ### Author Response · Authors · 2025-11-21
> **Responses to Reviewer ZfYy (1/2)**
>
> Dear **Reviewer ZfYy**:
>
> We sincerely thank you for your thoughtful feedback and valuable suggestions on our work. We are pleased to clarify and answer the questions and provide some additional experimental results.
>
> # For Weakness 1 & Question 2
>
> Thank you for your suggestion. We acknowledge that our initial presentation had some shortcomings, particularly the over-complexity of Figure 1 and the lack of clarity in our explanation of dLLM mechanisms. **We have followed your advice and have revised our paper, to provide a clearer explanation and make it more accessible for readers who are not familiar with dLLMs.**
>
> # For Weakness 2 & Question 1
>
> Thank you for your observation. We agree that AR LLMs also exhibit some similar safety issues. For instance, [1] showed that decoding strategies significantly impact AR LLM safety, [2] revealed that dangerous information still exists in the top-k tokens of aligned models, and [3] found that LLM safety alignment might be superficial, existing only in the initial few tokens. We will revise our paper to include these necessary citations.
>
> However, we believe there is a fundamental distinction between these prior works and our own, stemming from a fundamental difference in the underlying context and the granularity of investigation.
>
> - Regarding [1], the decoding strategies for AR LLMs primarily concern the **token-level** sampling process (like temperature, top-p, and top-k). In contrast, our paper's investigation of decoding strategies focuses on the **intra-step remasking** policy, a mechanism unique to dLLMs. We did not focus on the token-level generation strategies. As for temperature adjustment, we found that simply altering the temperature is completely ineffective against jailbreak attacks. You can refer to *Multi-sampling attack* (Appendix D.3) for the detailed results.
> - Regarding [2], that work revealed how to manipulate probability distributions to jailbreak models. However, their focus was on how to sample from a **single token**'s probability distribution to compromise safety. Our work, conversely, focuses on the parallel decoding of **multiple token positions** within a single dLLM step, and we investigate the **mutual influence** of the probability distributions across these multiple positions.
> - Regarding [3], it was shown that a model's safety alignment might be shallow, existing superficially only in the early output tokens. We acknowledge the similarity to our Denoising-Path Dependence, as both are fundamentally related to conditional probability in model inference (Section 3.3). However, we argue that this characteristic is more pronounced in dLLMs compared to AR LLMs. This is partly because dLLMs have a **non-fixed generation order**, compelling an analysis of the entire process from a step-level perspective. Furthermore, dLLMs utilize bidirectional attention, which offers greater potential for integrating global knowledge.
>
> We appreciate your in-depth analysis of our points. **We have revised the paper to include the more detailed discussion above, clarifying the unique aspects of our findings compared to similar phenomena in AR LLMs.**
>
> [1] Huang, Yangsibo, et al. "Catastrophic jailbreak of open-source llms via exploiting generation." arXiv preprint arXiv:2310.06987 (2023).
>
> [2] Zhang, Zhuo, et al. "On large language models’ resilience to coercive interrogation." 2024 IEEE Symposium on Security and Privacy (SP). IEEE, 2024.
>
> [3] Qi, Xiangyu, et al. "Safety alignment should be made more than just a few tokens deep." arXiv preprint arXiv:2406.05946 (2024).

---

> ### Author Response · Authors · 2025-11-21
> **Responses to Reviewer ZfYy (2/2)**
>
> # For Weakness 3 & Question 3
>
> Thank you for your valid point. To the best of our knowledge, no existing work has studied adaptive attacks specifically targeting dLLMs. However, we agree with your assessment that including an evaluation against adaptive attacks would make the DiffuGuard framework more persuasive.
>
> We designed and implemented three novel adaptive attack scenarios for supplementary experiments:
>
> - **Multi-sampling attack** that attempts to jailbreak by repeatedly sampling each query using diverse methods, assuming the attacker is aware of the stochastic remasking defense;
> - **Gradient-based optimization attack** on Safety Divergence (SD), assuming the attacker knows how SD is calculated and used for detection;
> - **Heuristic threshold-probing attack**, which also assumes knowledge of the SD threshold-based detection.
>
> We conducted additional experiments to evaluate DiffuGuard's performance under these attacks, as shown in the table below.
>
> | ASR(%)     | PAD$_\text{AdvBench}$ | + Multi-sampling Attack | + Gradient-based Attack (on SD) | + Heuristic threshold-probing Attack (on SD) |
> | ---------- | --------------------- | ----------------------- | ------------------------------- | -------------------------------------------- |
> | LLaDA-8B   | 93.65                 | 98.85                   | 88.27                           | 98.84                                        |
> | +DiffuGuard | 24.42                 | 35.96                   | 44.42                           | 65.58                                        |
>
> As can be seen, DiffuGuard continues to demonstrate strong security performance even under these adaptive attacks, proving its effectiveness in adversarial settings. **We have incorporated them into the final paper (Appendix D.3) to make the content more comprehensive.**
>
> ---
>
> **We sincerely look forward to further discussion to clarify any other concerns. If you are satisfied with our reply and the revisions, we kindly hope you will consider increasing our score.**

---

> ### Author Response · Authors · 2025-11-27
> **Thank you & Looking forward to further discussion**
>
> Dear **Reviewer ZfYy**,
>
> We deeply appreciate your dedication to engaging in author–reviewer discussions. To facilitate clearer communication, we summarize your key concerns and our corresponding clarifications below:
>
> 1. **Regarding the presentation (W1&Q2)**: We have revised the paper's presentation, particularly the background section (Section 2), to ensure that readers less familiar with dLLMs can accurately grasp the background and our core ideas.
> 2. **Regarding the novelty of the insights (W2&Q1)**: We have provided a detailed discussion regarding the relationship between your cited works' findings and our insights. Although superficially similar, these analyses fundamentally differ due to the unique characteristics of dLLMs. We have updated the paper with this discussion (Appendix F.3).
> 3. **Regarding the adaptive attacks (W3&Q3)**: We conducted additional experiments on the adaptive attacks you mentioned, where attackers are assumed to be aware of our defense mechanism and optimize their strategies accordingly. We found that DiffuGuard still exhibits strong performance against these attacks. We have updated the paper with these results (Appendix D.3).
>
> For other issues not mentioned here, please refer to our detailed rebuttal response. We sincerely hope this addresses your concerns. Looking forward to further discussion with you!
>
> Warm regards,
>
> The Authors

---

### Official Review · Reviewer_1icH · 2025-10-31

**Soundness:** 4
**Presentation:** 3
**Contribution:** 3
**Rating:** 6
**Confidence:** 2

**Summary:**

This paper presents DIFFUGUARD, a training-free and plug-and-play defense framework designed to enhance the safety of Diffusion Large Language Models (dLLMs). Unlike traditional autoregressive LLMs, dLLMs generate text through iterative refinement and parallel decoding, which introduces unique vulnerabilities to jailbreak attacks. Through a dual-perspective safety analysis—intra-step (within-step dynamics) and inter-step (across-step dynamics), the authors identify two key weaknesses: the greedy Low Confidence Remask strategy that favors harmful tokens, and the Denoising-path Dependence, where early unsafe tokens can steer the entire generation toward unsafe outputs. DIFFUGUARD addresses these issues via two modules: Stochastic Annealing Remasking, which introduces adaptive randomness to mitigate greedy bias, and Block-level Audit and Repair, which detects and corrects unsafe content using internal model representations. Experiments across four dLLMs and multiple attack datasets show that DIFFUGUARD significantly reduces the Attack Success Rate (from 47.9% to 14.7%) while maintaining model performance and efficiency. The framework provides a robust, efficient, and generalizable solution for improving the safety of diffusion-based language models.

**Strengths:**

* The paper provides the first systematic safety analysis of Diffusion Large Language Models (dLLMs), a new and fast-growing generation paradigm distinct from autoregressive LLMs.
* The paper presents a well-structured analytical framework, decomposing safety evaluation into intra-step and inter-step dynamics, which is both theoretically sound and empirically verifiable.

**Weaknesses:**

The threat model is oversimplified and lacks depth.

It remains unclear how DIFFUGUARD would behave under adaptive attackers who are aware of the defense mechanism. Without such analysis, the framework’s robustness against dynamic or evolving attack strategies is uncertain.

Although DIFFUGUARD is presented as model-agnostic, its evaluation is limited to a small set of academic dLLMs (LLaDA, Dream, MMaDA, and LLaDA-1.5)

The safety evaluation mainly focuses on jailbreak-style text attacks.
Other important safety dimensions—such as misinformation, bias propagation, or multi-turn adversarial interactions—are not tested.

**Questions:**

* The paper assumes a partial white-box attacker. How would DIFFUGUARD perform under a black-box or adaptive attack setting?

* The experiments focus mainly on jailbreak attacks. Have the authors tested DIFFUGUARD on other safety aspects such as bias or factuality robustness?

* Does DIFFUGUARD ever over-defend (i.e., mistakenly censor harmless content)? If so, how often and under what conditions?

* Could DIFFUGUARD be applied during fine-tuning or reinforcement learning phases as a safety regularizer rather than a post-hoc inference module?

---

> ### Author Response · Authors · 2025-11-21
> **Responses to Reviewer 1icH (1/2)**
>
> Dear **Reviewer 1icH**:
>
> We sincerely thank you for your thoughtful feedback and valuable suggestions on our work! We are pleased to clarify and answer these questions.
>
> # For Weakness 1
>
> Thank you for this valid point. We acknowledge that the description of our threat model in the initial submission was overly concise. **We have revised this section in the paper to make it more detailed and clear, explicitly defining the capabilities and goals of both the attacker and defender.**
>
> However, we believe the underlying model itself is valid and appropriate, as it reflects common and practical attack-defense scenarios[1][2]. Therefore, we consider it a reasonable foundation for our work.
>
> [1] Zou, Andy, et al. "Universal and transferable adversarial attacks on aligned language models." *arXiv preprint arXiv:2307.15043* (2023).
>
> [2] Liu, Xiaogeng, et al. "Autodan: Generating stealthy jailbreak prompts on aligned large language models." *arXiv preprint arXiv:2310.04451* (2023).
>
> # For Weakness 2
>
> Your concern is reasonable. Although our initial objective was to defend against several common jailbreak attacks, evaluating DiffuGuard's performance under adaptive attackers is indeed important.
>
> As there are currently no established adaptive attack methods for dLLMs, we defined and implemented three novel adaptive attack strategies:
>
> - **Multi-sampling attack** that attempts to jailbreak by repeatedly sampling each query using diverse methods, assuming the attacker is aware of the stochastic remasking defense;
> - **Gradient-based optimization attack** on Safety Divergence (SD), assuming the attacker knows how SD is calculated and used for detection;
> - **Heuristic threshold-probing attack**, which also assumes knowledge of the SD threshold-based detection.
>
> We conducted additional experiments to evaluate DiffuGuard's performance under these attacks, as shown in the table below.
>
> | ASR(%)     | PAD$_\text{AdvBench}$ | + Multi-sampling Attack | + Gradient-based Attack (on SD) | + Heuristic threshold-probing Attack (on SD) |
> | ---------- | --------------------- | ----------------------- | ------------------------------- | -------------------------------------------- |
> | LLaDA-8B   | 93.65                 | 98.85                   | 88.27                           | 98.84                                        |
> | +DiffuGuard | 24.42                 | 35.96                   | 44.42                           | 65.58                                        |
>
> As can be seen, DiffuGuard continues to demonstrate strong security performance even under these adaptive attacks, proving its effectiveness in adversarial settings. **We have added the detailed experimental design and results to the paper.**
>
> # For Weakness 3
>
> Thank you for your suggestion.
>
> First, our defender operates in a *white-box setting*. DiffuGuard requires access to the model's logits and the ability to modify the inference process. The few industrial dLLMs (Mercury Coder or Gemini Diffusion) have not open-sourced their weights. Therefore, it is not feasible to apply our DiffuGuard framework to them.
>
> Second, the dLLM field is still in its early stages but is rapidly evolving, and there are *not many industrial dLLMs available for evaluation*. Consequently, we selected the two most popular open-source dLLM architectures (LLaDA and Dream) and evaluated four representative models based on them to ensure the transferability of our conclusions. The vast majority of other academic dLLMs are based on these two paradigms. Therefore, we believe that evaluating these four representative models is a reasonable and generalizable approach.
>
> # For Weakness 4
>
> Thank you for your insight. As defined in our Threat Model, our work specifically focuses on the safety of dLLMs within the context of jailbreak attacks. We acknowledge that this is an initial exploration of a new and emerging topic. Our contribution is to present key insights into dLLM-specific vulnerabilities and provide a targeted strategy for this defined threat. We agree that other important safety dimensions, as you mentioned, are valuable topics. We plan to explore these broader safety topics in our future work.

---

> ### Author Response · Authors · 2025-11-21
> **Responses to Reviewer 1icH (2/2)**
>
> # For Question 1
>
> Thank you for your insight.
>
> In Section 5.2, we use WildJailbreak and JBB-Behaviors datasets, which contain pre-optimized jailbreak prompts, and we use these prompts to directly query the model. We consider these to be black-box attacks. This is in contrast to other methods like PAD, DIJA, and AutoDAN, which follow the partial white-box attack configuration.
>
> As for the adaptive attack setting, we have provided supplementary experiments in our response to your Weakness 2. These experiments demonstrate that DiffuGuard maintains strong performance even in this scenario.
>
> # For Question 2
>
> While we did not conduct tests specifically targeting these domains, our experiments indirectly touch upon these aspects. The datasets we employed (WildJailbreak and JBB-Behaviors) already include a substantial amount of content related to bias and factuality robustness.
>
> We agree that testing against these specific safety sub-tasks is a valuable research direction, and we plan to explore DiffuGuard's performance on these topics in future work.
>
> # For Question 3
>
> Thank you for your observation. We agree that a robust defense framework must avoid over-defense (or, false positives) as much as possible.
>
> In Figure 6, our "Acc@Safe Query" metric is specifically designed to evaluate this. We found that on both LLaDA and Dream, DiffuGuard did not have a significant negative impact on harmless requests. This demonstrates that DiffuGuard does not compromise the model's general-purpose capabilities. A more detailed description of how the Safe Queries were constructed can be found in Appendix B.2.
>
> # For Question 4
>
> Thank you for your insightful suggestion; that is a very novel idea. We agree that DiffuGuard could indeed be used for such a purpose.
>
> For instance, during the RL phase for dLLMs, DiffuGuard could be applied during the sample rollout stage to regulate the safety of the model's outputs. This would help in generating higher-quality and safer inference data for the policy update, without significantly compromising the data distribution. We will certainly follow up on this promising direction in our future work to expand DiffuGuard's applications and enhance the inherent safety of dLLMs.
>
> ---
>
> **We sincerely look forward to further discussion to clarify any other concerns. If you have additional suggestions, we are committed to addressing them and continuously improving our work based on your feedback.**

---

### Official Review · Reviewer_TWn4 · 2025-11-01

**Soundness:** 3
**Presentation:** 4
**Contribution:** 4
**Rating:** 8
**Confidence:** 3

**Summary:**

The paper investigates safety vulnerabilities in Diffusion LLMs, which differ from autoregressive LLMs due to their iterative and parallel generation mechanisms. The authors conduct an in-depth analysis decomposed into two dimensions: intra-step dynamics (parallel generation within a single step) and inter-step dynamics (iterative refinement across steps). To address identified vulnerabilities, the authors propose a DiffuGuard defense.

**Strengths:**

- The paper is very vlear and well-written, and presents the analysis of dLLM vulnerabilities, which is very timely and addresses an underexplored area. As these models gain traction, understanding their unique vulnerabilities becomes important. This work provides valuable exploration of dLLM-specific attack surfaces that differ from traditional autoregressive LLMs.

- The experimental evaluation is comprehensive. The paper evaluates four different dLLMs across multiple datasets and uses diverse jailbreak methods suited for both AR LLMs and dLLMs. Authors use multiple evaluation metrics, considere utility trade-offs and baseline defenses. Overall, the paper seems thorough and well-executed.

- The paper introduces DiffuGuard and thoroughly investigates how it affects model safety and utility. The ablation studies clearly demonstrate the contribution of each component.

**Weaknesses:**

- My main concern is the lack of adaptive attacks. The paper evaluates against existing jailbreak methods (some designed for AR LLMs, others like PAD and DIJA specifically for dLLMs), but doesn't consider adaptive attacks where adversaries know about DiffuGuard's defense mechanisms. Given that the defense uses stochasticity and cosine distance thresholds, an adaptive attacker could potentially craft attacks that account for stochastic remasking, optimize prompts to minimize the Safety Divergence metric, or design attacks that bypass the block-level audit. I understand this is initial exploration focused on intrinsic vulnerabilities, but the lack of adaptive attack evaluation makes it hard to assess how robust DiffuGuard actually is in adversarial settings.

- The teaser figure (Figure 1) is extremely hard to parse and could be simplified for better readability.

**Questions:**

1. Have you considered evaluating against adaptive attacks where the adversary is aware of DiffuGuard's mechanisms? Given that DiffuGuard does not bring ASR to 0, I am concerned that the simplest adaptive attacks would revert the gains inroduced by the defense.

2. I would appreciate more analysis on how sensitive the defense is to hyperparameters and whether they transfer between different models. Currently, the paper shows hyperparameter settings for each model (Table 6), but it's unclear how to set, e.g. paprameter alpha for new models.

---

> ### Author Response · Authors · 2025-11-21
> **Responses to Reviewer TWn4 (1/1)**
>
> Dear **Reviewer TWn4**:
>
> We sincerely thank you for your thoughtful feedback and valuable suggestions on our work!
>
> We are pleased to clarify and answer the questions and provide some additional experimental results.
>
> # For Weakness 1 & Question 1
>
> Thank you for this valid point. As you noted, the primary focus of this paper is on the intrinsic vulnerabilities of dLLMs, and our evaluation covered several general and dLLM-specific jailbreak datasets and methods.
>
> Nevertheless, we agree with your assessment that including an evaluation against adaptive attacks would make DiffuGuard more persuasive. To the best of our knowledge, no existing work has studied adaptive attacks targeting dLLMs. Therefore, we designed and implemented three adaptive attack scenarios for supplementary experiments:
>
> - **Multi-sampling attack** that attempts to jailbreak by repeatedly sampling each query using diverse methods, assuming the attacker is aware of the stochastic remasking defense;
> - **Gradient-based optimization attack** on Safety Divergence (SD), assuming the attacker knows how SD is calculated and used for detection;
> - **Heuristic threshold-probing attack**, which also assumes knowledge of the SD threshold-based detection.
>
> We conducted additional experiments to evaluate DiffuGuard's performance under these attacks, as shown in the table below.
>
> | ASR(%)     | PAD$_\text{AdvBench}$ | + Multi-sampling Attack | + Gradient-based Attack (on SD) | + Heuristic threshold-probing Attack (on SD) |
> | ---------- | --------------------- | ----------------------- | ------------------------------- | -------------------------------------------- |
> | LLaDA-8B   | 93.65                 | 98.85                   | 88.27                           | 98.84                                        |
> | +DiffuGuard | 24.42                 | 35.96                   | 44.42                           | 65.58                                        |
>
> As shown, DiffuGuard continues to exhibit strong safety performance even when the attacker is aware of its underlying mechanisms, demonstrating its effectiveness in adversarial settings.
>
> **We have added the detailed experimental design and results of these adaptive attacks to the paper (Appendix D.3).** Thank you again for your constructive suggestion.
>
> # For Weakness 2
>
> Thank you for your suggestion. We will revise this figure by removing unnecessary and redundant elements and modifying the parts that are difficult to understand, making it clearer and more readable.
>
> **We have already updated the new teaser figure in the revised PDF.**
>
> # For Question 2
>
> Thank you for your valuable comment. We agree that analyzing hyperparameter sensitivity and their transferability is crucial. We have conducted experiments to address this, which are detailed in Appendix D (Tables 7 and 8). In these tables, we vary one hyperparameter at a time while keeping the others fixed (at their default values from Table 6), observing the impact on both ASR and general capability.
>
> We offer the following recommendations:
>
> 1. $\alpha_0$: As discussed in Section 3.2, a high $\alpha_0$ enhances safety but potentially degrades general capability. For new models, we recommend starting at 0.5 and tuning it to achieve the desired balance between safety and utility.
> 2. $\lambda$ and $\gamma$: These parameters function together in the Block-level Audit and Repair module. $\lambda$ is a model-specific parameter, as different models' hidden states exhibit varying sensitivities to cosine similarity. $\gamma$ also represents a trade-off between safety and general capability.
> 3. extra_steps: We set this to 8, which we found to be an efficient value that does not significantly impair model utility.
>
> In the main experiments of the paper, we selected a hyperparameter configuration for each model that demonstrated a balanced performance.
>
> ---
>
> **We sincerely look forward to further discussion to clarify any other concerns.**

---

> > ### Comment · Reviewer_TWn4 · 2025-11-21
> >
> > Dear Authors,
> >
> > Thank you for your rebuttal.
> >
> > After carefully reading other reviews, I must emphasize that I do share concerns about sufficient adaptive evaluations and overclaiming the robustness of the proposed mitigations in the same capacity as other reviewers. However, given that the paper in many regards pioneers the direction of dLLMs safety, sheds light on how it is different from AR LLMs and otherwise contains valuable insights, I find the proposed adaptive red teaming sufficient.
> >
> > I will maintain my score and recommend acceptance.

---

> > > ### Author Response · Authors · 2025-11-24
> > >
> > > Dear Reviewer TWn4,
> > >
> > > Thank you for your positive feedback and for recognizing the contributions of our work—it means a lot to us and inspires us to continue improving.
> > >
> > > We are sincerely grateful for the time and expertise you have shared, and we look forward to any further questions you might have.
> > >
> > > Best regards,
> > >
> > > The Authors

---

### Author Response · Authors · 2025-11-29
**Rebuttal Summary for ACs and SACs (1/5)**

Dear **Area Chairs**, **Senior Area Chairs**, and **Reviewers**,

We are deeply grateful for your efforts concerning the OpenReview leak incident. We understand the demands on your time and deeply appreciate your willingness to review our paper and provide a fair assessment.

To help Area Chairs clearly understand the discussions during the rebuttal, we have summarized the Strengths raised by the reviewers, and the content and outcomes of our discussions with each reviewer regarding the Weaknesses and Questions, and submit this summary for your reference. We have uploaded the revised version of the manuscript, and all changes are highlighted in blue.

# Strengths

| Strengths                                 | Details                                                      |
| ----------------------------------------- | ------------------------------------------------------------ |
| **Timely and Critical Topic**             | **TWn4:** "The paper is... **very timely and addresses an underexplored area**. As these models gain traction, understanding their unique vulnerabilities becomes important." |
|                                           | **1icH:** "The paper provides the **first systematic safety analysis** of Diffusion Large Language Models (dLLMs), a new and fast-growing generation paradigm..." |
|                                           | **ZfYy:** "The paper investigates a critical safety issue in diffusion-based LLMs, an **emerging and underexplored direction**." |
|                                           | **3R1N:** "...tackles the problem of safety in Diffusion Large Language Models, which I think is a **relevant and underexplored area** compared to autoregressive models." |
| **Insightful Analytical Framework**       | **1icH:** "The paper presents a **well-structured analytical framework**, decomposing safety evaluation into intra-step and inter-step dynamics, which is both **theoretically sound and empirically verifiable**." |
|                                           | **oL9w:** "Section 3 offers a good perspective of dLLMs' security issues: the analysis of the model’s decoding mechanism vulnerabilities is **clear and well presented**." |
|                                           | **3R1N:** "The authors frame this analysis around 'intra-step' and 'inter-step' dynamics, which is a **conceptualization that helps to structure the problem**." |
| **Logical Method Design**                 | **ZfYy:** "The approach’s motivation is **well-supported by in-depth analysis** of dLLM internals." |
|                                           | **3R1N:** "The proposed defense... is **directly motivated by the preceding analysis**... I find this **link between the problem analysis and the solution design to be logical**." |
|                                           | **TWn4:** "The ablation studies **clearly demonstrate the contribution** of each component." |
| **Comprehensive Experimental Evaluation** | **TWn4:** "The **experimental evaluation is comprehensive**. The paper evaluates four different dLLMs across multiple datasets... considers utility trade-offs and baseline defenses." |
|                                           | **ZfYy:** "The evaluation **demonstrates the effectiveness** of DiffuGuard against state-of-the-art jailbreak attacks." |
|                                           | **oL9w:** "The authors evaluate the defense against **dLLM-specific jailbreak methods**." |
|                                           | **3R1N:** "The experimental validation **covers four dLLMs and six different attack methods**... The authors also include experiments on model utility and inference speed..." |
| **Clear Presentation**                    | **TWn4:** "The paper is **very clear and well-written**..."  |
|                                           | **3R1N:** "The paper is generally **clearly written**, and figures like Figure 1 are helpful..." |

---

> ### Author Response · Authors · 2025-11-29
> **Rebuttal Summary for ACs and SACs (2/5)**
>
> # Weaknesses&Questions
>
> | Reviewer | Comments                                                     | Our Response                                                 | Rating(Confidence) | Reviewer Response                                            |
> | -------- | ------------------------------------------------------------ | ------------------------------------------------------------ | ------------------ | ------------------------------------------------------------ |
> | TWn4     | "My main concern is the lack of **adaptive attacks**." `(W1&Q1)` | We supplemented the **detailed adaptive attack experiments** `(Appendix D.3)`, which have been positively acknowledged by the reviewer. Experimental results indicate that DiffuGuard also exhibits stable defense performance against these adaptive attacks. | **8(3)**           | Solved; Maintain the positive score and recommend acceptance. |
> |          | "The teaser figure (**Figure 1**) is extremely hard to parse and could be simplified for better readability." `(W2)` | We have **revised `Figure 1`** to enhance its clarity.       |                    |                                                              |
> |          | "I would appreciate more analysis on how **sensitive** the defense is to **hyperparameters** and whether they transfer between different models." `(Q2)` | We have **analyzed the sensitivity of the hyperparameters** in detail in the initial draft of the paper `(Appendix D, Tables 7 and 8)`, and supplemented this with an **analysis of the parameter values and transferability**. |                    |                                                              |
> | 1icH     | "The **threat model** is oversimplified and lacks depth." `(W1)` | We maintain that our threat model represents a **generally accepted configuration** within the field of LLM safety and does not lack depth. Following the reviewer's suggestion, we have **revised the "Threat Model" section** `(Section 3.1)` to be more detailed and clear. | **6(2)**           | No response                                                  |
> |          | "It remains unclear how DIFFUGUARD would behave under **adaptive attackers** who are aware of the defense mechanism." `(W2&Q1)` | We supplemented the **detailed adaptive attack experiments** `(Appendix D.3)`, which have been positively acknowledged by the Reviewer TWn4. Experimental results indicate that DiffuGuard also exhibits stable defense performance against these adaptive attacks. |                    |                                                              |
> |          | "Although DIFFUGUARD is presented as model-agnostic, its evaluation is **limited to a small set of academic dLLMs**." `(W3)` | **We respectfully disagree with the reviewer and have clarified the constraints of the current dLLM landscape.** (1) DiffuGuard requires white-box access, while most industrial dLLMs are not open-source; (2) there are currently few industrial dLLMs available; (3) we follow the practice of most representative dLLM works by utilizing the two most mainstream architectures. |                    |                                                              |
> |          | "Other important **safety dimensions**—such as misinformation, bias propagation, or multi-turn adversarial interactions—are not tested." `(W4&Q2)` | **We have explained the reasons to the reviewers.** (1) According to our Threat Model, our work specifically focuses on the safety of dLLMs within the context of jailbreak attacks; (2) while we did not conduct tests specifically targeting these domains, our experiments (using WildJailbreak and JBB-Behaviors) indirectly touch upon these aspects. |                    |                                                              |
> |          | "Does DIFFUGUARD ever **over-defend** (i.e., mistakenly censor harmless content)? If so, how often and under what conditions?" `(Q3)` | We had **already investigated this issue** in the initial version of the paper. Our "Acc@Safe Query" metric is specifically designed to evaluate this `(Section 5.3, Figure 6)`. We found that DiffuGuard did not have a negative impact on harmless requests. This demonstrates that DiffuGuard does not compromise the model's general-purpose capabilities. |                    |                                                              |
> |          | "Could DIFFUGUARD be applied during fine-tuning or reinforcement learning phases as a **safety regularizer** rather than a post-hoc inference module?" `(Q4)` | We thank the reviewer for their suggestion regarding our future work. Utilizing DiffuGuard as a safety regularizer during the LLM post-training phase is an interesting idea that we plan to explore further. |                    |                                                              |

---

> > ### Author Response · Authors · 2025-11-29
> > **Rebuttal Summary for ACs and SACs (3/5)**
> >
> > | **Reviewer** | Comments                                                     | Our Response                                                 | Rating(Confidence) | Reviewer Response                                            |
> > | ------------ | ------------------------------------------------------------ | ------------------------------------------------------------ | ------------------ | ------------------------------------------------------------ |
> > | ZfYy         | "The presentation could be improved; a more intuitive **explanation of diffusion LLM mechanisms** would help non-experts better grasp the core idea." `(W1&Q2)` | We have followed the reviewer's advice and have **revised our paper** `(Section 2)`, to provide a clearer explanation and make it more accessible for readers who are not familiar with dLLMs. | **4(3)**           | No response                                                  |
> > |              | "The key observations may **not be entirely novel** and appear to manifest in AR LLMs as well." `(W2&Q1)` | We respectfully disagree with the reviewer's assessment regarding the novelty of our paper. We have conducted a detailed analysis of the relationship and differences between DiffuGuard and the papers cited by the reviewer, and **elaborated on the unique novelty points** of our framework `(Appendix F.3)`. These analyses are unique and **represent a previously unexplored area of research**, a point corroborated by Reviewer TWn4, who stated: "...the paper in many regards pioneers the direction of dLLMs safety, sheds light on how it is different from AR LLMs and otherwise contains valuable insights..." |                    |                                                              |
> > |              | "There is limited discussion of potential **adaptive attacks**." `(W3&Q3)` | We supplemented the **detailed adaptive attack experiments** `(Appendix D.3)`. Experimental results indicate that DiffuGuard also exhibits stable defense performance against these adaptive attacks. |                    |                                                              |
> > | oL9w         | "There is no guarantee that the model is actually aligned to reject the original malicious query, meaning that the baseline representation of *p_origin* **may not correspond to a genuine refusal state**." `(W1&Q1)` | We believe that relying on the model's inherent harmfulness refusal capability is reasonable and have provided evidence to support our assumption. We conducted **detailed experiments** demonstrating that various dLLMs exhibit **a perfect refusal rate** against malicious queries `(Figure 4&10)`, which validates this assumption under our Threat Model. | **2(4)**           | Participated in further discussions, but have not yet responded. |
> > |              | "The paper does not include an ablation across diverse jailbreak algorithms such as AutoDAN" `(Q1)` | In response to the reviewer's suggestion, we conducted **additional ablation studies** covering other jailbreak algorithms `(Section 5.4, Table 2)`. |                    |                                                              |
> > |              | "What I'm worried about is what happens when this assumption does not hold: what if we want to protect a model that is **not fully aligned to refuse the initial malicious query**? Will DiffuGuard fail in this kind of scenario?" `(Extra W1)` | We respectfully disagree with the reviewer's view on "not fully aligned" models. (1) **We do not believe they constitutes a practically relevant scenario.** We do not consider models that have received no safety adjustments to be within the scope of practical application, as discussing their safety **has limited practical utility**; (2) should a scenario necessitate defending a not fully aligned dLLM, our Stochastic Annealing Remasking module can **still provide a degree of defense**. |                    |                                                              |
> > |              | "The information captured by SD **varies across jailbreak types**, which may reduce the reliability and effectiveness of the defense." `(W1&Q1)` | We agree with the reviewer's points regarding "different jailbreak methods may steer the input representation towards different clusters". However, we argue that **[1] actually supports our approach**. Figure 7 in [1] demonstrates that all effective jailbreak attacks function by suppressing the 'harmfulness' feature, and SD is designed to identify a jailbreak attempt by measuring the significant deviation from the inherent 'harmfulness' representation that occurs when a template forces the model to masquerade as benign. |                    |                                                              |

---

> > > ### Author Response · Authors · 2025-11-29
> > > **Rebuttal Summary for ACs and SACs (4/5)**
> > >
> > > | **Reviewer** | Comments                                                     | Our Response                                                 | Rating(Confidence) | Reviewer Response |
> > > | ------------ | ------------------------------------------------------------ | ------------------------------------------------------------ | ------------------ | ----------------- |
> > > | oL9w         | "I would therefore suggest that the authors **employ more effective state-of-the-art attack techniques**, which would provide a more complete and reliable assessment of DiffuGuard’s robustness." `(W2&Q2)` | We have adopted reviewer's suggestion to **re-evaluate the Attack Success Rate using HarmBench and StrongReject** `(Appendix D.5, Table 11)`. We also supplemented the **detailed adaptive attack experiments** `(Appendix D.3)`. Experimental results indicate that DiffuGuard also exhibits stable defense performance against these adaptive attacks. |                    |                   |
> > > |              | "I still do not understand whether there is a reason why you considered GCG jailbreaks against Llama3-8B-Instruct in your evaluation or not." `(Extra W2)` | We have provided the reviewers with a detailed explanation of our reason for this choice. As of the conference submission deadline, no effective GCG method specifically targeting dLLMs was available, and the conventional GCG algorithm is difficult to optimize due to the differences between dLLM and AR LLM generation paradigms. Therefore, in our experiments, we directly transferred the pre-optimized GCG suffixes from the LLaMA model, leveraging their architectural similarity [2] and the known transferability of the GCG algorithm, as detailed in Appendix B.2 in [3]. We acknowledged the limitation of this transfer-based setup due to the lack of dLLM-specific GCG attacks at the time. Consequently, we supplemented our evaluation with an **effective gradient-based attack experiment** `(Appendix D.3)`, and the results show that DiffuGuard still maintains good defense efficacy against this type of attack. |                    |                   |
> > > |              | "I believe the values you reported underline the limitations of the setup you used to evaluate the effectiveness of DiffuGuard." `(Extra W2)` | There was a discrepancy in the interpretation of the results we presented. **We found that this misunderstanding stemmed from an issue with the HarmBench evaluation procedure and initial configuration issues, which have now been fully corrected.** Following correction, we obtained precise evaluation results, which have been shared with the reviewer. We believe these new results are sufficient to address the concern, as the **outcomes on these transferable benchmarks further confirm DiffuGuard's effectiveness**. However, the reviewer has not yet responded to our latest reply. |                    |                   |
> > > |              | "I consider **the use of GPT-4o-mini as a strong limitation** of the authors’ evaluation setup, as it does not provide a robust or standardized measure of the defense’s effectiveness." `(Q3)` | We have added **additional evaluations** using: (1) LLaMA Guard, and (2) Human evaluation `(Appendix D.5, Table 10)`. The results show that LLaMA Guard exhibits similar trends to GPT-4o-mini assessment. Furthermore, our GPT-4o-mini assessment demonstrates strong alignment with the Human Expert evaluation. |                    |                   |
> > > |              | "I have an additional question regarding the human evaluation. How many people were involved in the labelling of the model's responses?" `(Extra W3)` | We have provided the reviewer with the **detailed number** of participants, the specific tasks, and the complete protocol for our human evaluation. |                    |                   |
> > > |              | "Although the analysis in Section 3 is insightful, the authors base both their observations and conclusions solely on results **from a single model**." `(Q4)` | Following the reviewer's suggestion, we have **updated our paper** to include our analysis on other dLLMs in `Appendix D.4`. |                    |                   |
> > >
> > > [1] Ball, Sarah, Frauke Kreuter, and Nina Panickssery. "Understanding jailbreak success: A study of latent space dynamics in large language models." *arXiv preprint arXiv:2406.09289* (2024).
> > >
> > > [2] Nie, Shen, et al. "Large language diffusion models." *arXiv preprint arXiv:2502.09992* (2025).
> > >
> > > [3] Zou, Andy, et al. "Universal and transferable adversarial attacks on aligned language models." *arXiv preprint arXiv:2307.15043* (2023).

---

> > > > ### Author Response · Authors · 2025-11-29
> > > > **Rebuttal Summary for ACs and SACs (5/5)**
> > > >
> > > > | **Reviewer** | Comments                                                     | Our Response                                                 | Rating(Confidence)                   | Reviewer Response |
> > > > | ------------ | ------------------------------------------------------------ | ------------------------------------------------------------ | ------------------------------------ | ----------------- |
> > > > | oL9w         | "In Figure 1, some elements are difficult to read, especially on the right side." `(Minors)` | We have **optimized `Figure 1`** to make it easier to read.  |                                      |                   |
> > > > |              | "In Table 2, it would be useful to also report the contribution of each component of the defense on *safe queries*." `(Minors)` | We have added a **more fine-grained ablation study** to `Table 2`, targeting each component of the defense. |                                      |                   |
> > > > |              | "It would be useful to analyze also how much performance drops on safe queries." `(Minors)` | We have added the **impact of Random Remasking on safe queries** (we use GSM8K) to ablation study (Table 2). |                                      |                   |
> > > > |              | "In Figure 4, it would be interesting to include the effect of a jailbreak query with the prefilling token *“Sure”* to highlight differences compared to the *“Sorry”* prefilling." `(Minors)` | Based on the reviewer's suggestion, we conducted **additional experiments** and have updated `Figure 4` accordingly. We found that adding "Sure" to the jailbreak query increases the ASR to 94.7%. |                                      |                   |
> > > > | 3R1N         | "I would ask that you perform an evaluation against such an **adaptive attack**, as this would be critical for assessing the true robustness of DIFFUGUARD." `(W1&W2&Q1)` | We **explain and experimentally demonstrate** why white-box gradient descent methods targeting SD struggle to circumvent DiffuGuard's defense. We supplemented the **detailed adaptive attack experiments** `(Appendix D.3)`. Experimental results indicate that DiffuGuard also exhibits stable defense performance against these adaptive attacks. | **6(3)** -> **8(4)** (Before Nov 27) | No response       |
> > > > |              | "It's unclear how would be automatically extracted from more complex, natural-language jailbreaks" `(W3)` | PAD and DIJA are currently the dominant attack methods targeting dLLMs, and we believe DiffuGuard's contribution lies in **proposing the core methodology for detecting and defending against jailbreaks**. Furthermore, against potential future complex natural language jailbreaks, DiffuGuard can be readily improved—for instance, by using **rule-based** token segment splitting or by invoking an **LLM** for semantic analysis and decomposition. |                                      |                   |
> > > > |              | "For reproducibility, it would be helpful to know how these values (**hyperparameters**) were chosen and how to best tune them for new models." `(W3)` | We conducted a **detailed hyperparameter analysis** in `Appendix D`. We also explains the hyperparameter **selection process**, analyzes the **impact of key parameters**, and demonstrates the **best practices** for deployment. |                                      |                   |
> > > > |              | "It would be interesting to know if the authors believe these same vulnerabilities and the DIFFUGUARD defense would apply to **other dLLM architectures**, such as the uniform kenerls." `(W3)` | We explained to the reviewer why we focused our study on Masked Diffusion Model (MDM) and excluded other dLLM architectures. 'dLLM' is a broad term, and the underlying mechanisms between its specific implementations vary fundamentally. Our paper exclusively focuses on the MDM paradigm, as **it is currently the best-performing and most general dLLM architecture**. |                                      |                   |
> > > > |              | "A more **fine-grained ablation** on the sub-components would be even more convincing." `(W3)` | We have conducted more **fine-grained ablation experiments** targeting the sub-components within each module, as shown in `Table 2`. |                                      |                   |
> > > >
> > > > ---
> > > >
> > > > **We express our sincere gratitude to all reviewers for their constructive feedback. The suggestions and insights provided have been invaluable in improving our paper. We hope that this summary clarifies the questions raised by the reviewers.**
> > > >
> > > > **We sincerely thank ACs and SACs once more for your valuable time and insightful feedback. Your dedication contributes to making ICLR a more open, inclusive, fair, and equitable research community.**
> > > >
> > > > Best regards,
> > > >
> > > > The Authors

---

### Meta-Review · Area_Chair_vbDN · 2025-12-06

**Summary:**

Overall, the main concerns focused on three points. First, several reviewers emphasized the need for adaptive attack evaluation, as they considered this essential for validating the robustness of the defense.
Second, some reviewers raised questions about the novelty and clarity of the analytical contributions, particularly regarding whether the insights are unique to diffusion models.
Third, reviewers pointed to issues in the evaluation setup, including choice of attacks, consistency of metrics, and assumptions behind the block-level audit procedure.
Despite these points, there was strong positive feedback on the clarity of the analysis, the thoroughness of the experiments, and the importance of addressing safety in diffusion language models. The rebuttal specifically targeted the major concern around adaptive attacks and clarified misunderstandings regarding the audit mechanism and evaluation standards.

**Reviewer Concerns:**

The author rebuttal addressed the largest shared concern across reviewers, the absence of adaptive attack evaluation. The added experiments directly responded to this issue, and reviewer TWn4 confirmed they found the adaptive red teaming sufficient. This also resolves the corresponding concerns from reviewer 1icH and reviewer ZfYy, who were primarily aligned with the question of whether the defense generalizes beyond static attacks. Reviewer ZfYy concerns about novelty were also largely clarified by the rebuttal, which distinguished the diffusion mechanisms from autoregressive models and highlighted the unique contributions. For reviewer oL9w, many points were resolved through additional experiments with HarmBench, StrongREJECT, and guard models, and through clarification of the audit procedure. However, some skepticism from reviewer oL9w persists, particularly regarding the validity of the block-level audit and the classification of certain attacks. These remaining concerns appear tied to differences in evaluation philosophy rather than technical gaps.

**Reviewer Scores:**

Reviewer TWn4 already set their score high, so their rating likely remains 8->8.

Reviewer 1icH had concerns primarily about adaptive attacks, which the rebuttal addressed. 6->6.

Reviewer ZfYy didn't respond, concerns about novelty and adaptive attacks were directly engaged with and partially supported by another reviewer, so their score would likely rise modestly, 4->5.

Reviewer oL9w expressed the most persistent concerns, and although several misconceptions were clarified, likely staying 2->2.

---

### Decision · Program_Chairs · 2026-01-26

Accept (Poster)